# Research on Hand–Eye Calibration Accuracy Improvement Method Based on Iterative Closest Point Algorithm

Tingwu Yan [1] , Peijuan Li [2,*], Yiting Liu [3], Tong Jia [3], Hanqi Yu [2] and Guangming Chen [4]

1  College of Mechanical Engineering, Nanjing Institute of Technology, Nanjing 211167, China; mach_ytw@163.com
2  Industrial Center, College of Innovation and Entrepreneurship, Nanjing Institute of Technology, Nanjing 211167, China; yuhq@njit.edu.cn
3  College of Automation, Nanjing Institute of Technology, Nanjing 211167, China; 15195977181@163.com (Y.L.); jiatong@njit.edu.cn (T.J.)
4  College of Engineering, Nanjing Agricultural University, Nanjing 210031, China; 18252812244@163.com
*  Correspondence: y00450210433@njit.edu.cn

**Abstract:** In the functioning of the hand–eye collaboration of an apple picking robot, the accuracy of the hand–eye relationship is a key factor affecting the efficiency and accuracy of the robot's operation. In order to enhance the low accuracy of traditional hand–eye calibration methods, linear and nonlinear solving methods based on mathematical tools such as quaternions are commonly adopted. To solve the loss of accuracy in decoupling during the linearization solution and to reduce the cumulative error that occurs during nonlinear solutions, a hand–eye calibration method, based on the ICP algorithm, is proposed in this paper. The method initializes the ICP matching algorithm with a solution derived from Tsai–Lenz, and substitutes it for iterative computation, thereby ascertaining a precise hand–eye conversion relationship by optimizing the error threshold and iteration count in the ICP matching process. Experimental results demonstrate that the ICP-based hand–eye calibration optimization algorithm not only circumvents the issues pertaining to accuracy loss and significant errors during solving, but also enhances the rotation accuracy by 13.6% and the translation accuracy by 2.47% compared with the work presented by Tsai–Lenz.

**Keywords:** picking robots; hand–eye calibration; ICP matching algorithm; Tsai–Lenz

## 1. Introduction

Agricultural harvesting robots are electromechanical devices that integrate mechanization, automation, and intelligence. They have been researched and applied extensively in agricultural production. These robots are typically composed of components such as AGV automatic guided vehicles, end effectors, 3D vision cameras, and collaborative robotic arms [1–3]. During the harvesting process, the 3D vision camera perceives targets and extracts their pose information. Subsequently, after hand–eye calibration, the pose in the camera coordination frame is transformed into a pose relative to the robotic arm's base coordination frame [4]. The accuracy of hand–eye calibration is a crucial factor for the efficient operation of agricultural harvesting robots. The task of hand–eye calibration primarily involves solving for $X$ in $AX = XB$ to determine the hand–eye calibration relationship. In hand–eye calibration, the equation $AX = XB$ represents the transformation relationship between the camera frame and the robotic arm frame. $A$ and $B$ denote the transformation matrices of the camera frames and robotic arm frames relations, respectively. $X$ is the solution sought in the hand–eye calibration equation $AX = XB$. This equation delineates the translation and rotation relationship between the camera and robotic arm coordinate frames. Through solving this equation, precise hand–eye calibration results can be obtained, facilitating accurate spatial localization and control. Hence, the accuracy of $X$'s solution determines the precision of hand–eye calibration [5]. For the problem

of solving *X*, various scholars have employed different mathematical methods for the equation $AX = XB$. Common approaches include linear or nonlinear solving methods based on mathematical tools such as dual quaternions, rotation vectors, and so on [6–15]. The Tsai–Lenz method employs a nonlinear solving approach based on the least squares method for hand–eye calibration equations [16]. It employs a two-step process, initially determining the rotation matrix followed by implementing nonlinear computations for translational errors. Nonetheless, solving for the rotation matrix can potentially result in error accumulation, which subsequently propagates errors affecting translational accuracy throughout the procedure. Heikkila et al. [17] enhanced the approach through introducing a four-step calibration method; this approach surpasses the original two-step method. They directly employed a linear solving method to solve *R* and *T*, followed by applying Levenberg–Marquardt optimization *R* and *T* for further refinement. Daniilidis et al. [18] proposed a solution method based on dual quaternions. In comparison to the two-step approach, this method constructs a new set of linear equations using quaternions and then employs SVD (singular value decomposition) for the simultaneous solution of *R* and *T*. However, there are issues of coupled accuracy loss during the solving process. Qi et al. [19], building upon the dual quaternion algorithm, incorporated the characteristics of data on SO(4) to solve the problem. Compared to the dual quaternion algorithm, their method demonstrates improved stability and practicality. Van et al. [20] proposed a solving method based on the exponential formula product and linearly constrained singular value decomposition least squares algorithm to enhance calibration accuracy. With the progress in 3D camera and LiDAR sensor technologies, the utilization of 3D point cloud data in rigid registration has witnessed substantial advancement [21–23]. Point cloud registration involves finding a rigid transformation to align one point cloud as closely as possible with another, encompassing translation and rotation. The iterative closest point (ICP) algorithm and its variants are widely used methods for precise point cloud registration [24–28]. Zhao et al. [29] proposed a method for re-matching fractured rigid fractured surfaces in point clouds based on the ICP algorithm. Zhang et al. [30] introduced a 3D map creation method for motion estimation using a feature-based ICP algorithm with a discrete selection mechanism. Although the ICP matching algorithm is widely employed in point cloud registration, it demands accurate initial matches; otherwise, it may suffer from local convergence issues leading to substantial errors. To address these challenges, this paper presents an ICP iterative point cloud registration and hand–eye calibration method based on the Tsai–Lenz algorithm. This method employs the Tsai–Lenz algorithm to solve *X* in $AX = XB$; X facilitates the conversion of the calibration board's feature points' point cloud pose from the camera coordinate system to a pose relative to the robotic arm's base coordinates. Simultaneously, an ICP matching process is performed with the feature points' point cloud in the tool center point (TCP) frame of the robotic arm. *X* serves as the initial value for ICP matching. Upon completion of the matching process, the obtained rigid transformation becomes the refined hand–eye transformation.

## 2. Materials and Methods

### 2.1. Hand–Eye Calibration Model

According to the camera installation method, there are two approaches: "eye-in-hand" and "eye-to-hand" [31]. In the "eye-in-hand" scenario, the camera is mounted on the robotic arm's end effector and moves along with the robot's motion. In the "eye-to-hand" scenario, the camera remains fixed in position but can observe the robotic arm and the operating area. The "eye-to-hand" visual system maintains a fixed field of view, ensuring that target information is not lost when the robotic arm moves. The calibration objective in this scenario is to establish the transformation relationship between camera frame and robotic arm frame. This paper is based on this installation approach for modeling, as illustrated in Figure 1.

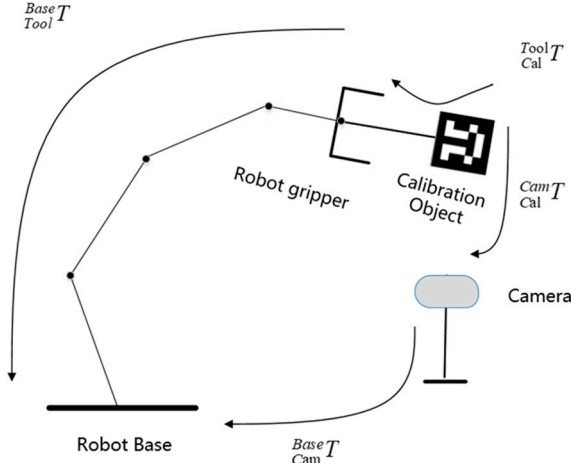

**Figure 1.** Hand–eye calibration model.

In Figure 1, $^{Tool}_{Cal}T$ represents the transformation relationship of the calibration board frame with respect to the end effector frame, $^{Cam}_{Cal}T$ represents the transformation relationship of the calibration board frame with respect to the camera frame, $^{Base}_{Cam}T$ represents the transformation relationship of the camera frame with respect to the robotic arm's base frame, and $^{Base}_{Cam}T$ represents the transformation relationship of the end effector frame with respect to the robotic arm's base frame. $^{Base}_{Cam}T$ signifies the hand–eye transformation relationship, which corresponds to $X$ in the hand–eye calibration equation $AX = XB$. During the grasping process of the harvesting robot, the 3D vision camera perceives and captures the pose of an apple. The imaging principle of a point $p(x_c, y_c, z_c)$ in the camera frame is illustrated in Figure 2.

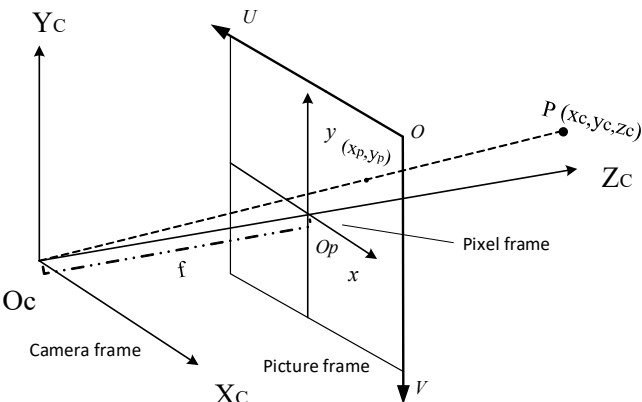

**Figure 2.** Camera imaging model.

Point $p(x_c, y_c, z_c)$ is imaged in the pixel coordinate frame $UoV$, and after undergoing the transformation in the image coordinate frame $xo_p y$ and subsequent projection, it is imaged in the camera coordinate frame $X_c o_c Y_c$. The transformation relationship is represented by Equation (1).

$$z_c \begin{bmatrix} u \\ v \\ 1 \end{bmatrix} = \begin{bmatrix} \frac{1}{dx} & 0 & u_0 \\ 0 & \frac{1}{dy} & v_0 \\ 0 & 0 & 1 \end{bmatrix} \cdot \begin{bmatrix} f & 0 & 0 \\ 0 & f & 0 \\ 0 & 0 & 1 \end{bmatrix} \cdot \begin{bmatrix} x_c \\ y_c \\ z_c \end{bmatrix} = \begin{bmatrix} \frac{f}{dx} & 0 & u_0 \\ 0 & \frac{f}{dy} & v_0 \\ 0 & 0 & 1 \end{bmatrix} \cdot \begin{bmatrix} x_c \\ y_c \\ z_c \end{bmatrix} = M_{in} \cdot \begin{bmatrix} x_c \\ y_c \\ z_c \end{bmatrix} \quad (1)$$

In Equation (1), $u_0, v_0$ represents the row and column of the object's imaged pattern in the pixel coordinate frame, $dx, dy$ represents the unit size length on the $x$ and $y$ axes for each pixel in the optical sensor, $f$ represents the camera focal length, and $M_{in}$ represents the camera intrinsic parameters, which can generally be obtained through the Zhang

calibration method. Hand–eye coordination tasks require transforming a point $p(x_c, y_c, z_c)$ in the camera coordinate frame into a point $p(x_r, y_r, z_r)$ relative to the robotic arm's base coordinate frame using the transformation relationship $_{Cam}^{Base}T$. This enables the robotic arm to perform harvesting operations. $_{Cam}^{Base}T$ can be represented using $3 \times 3$ rotation matrix $R$ and $3 \times 1$ translation vector $T$. This relationship is illustrated in Equation (2).

$$\begin{bmatrix} x_r \\ y_r \\ z_r \\ 1 \end{bmatrix} = \begin{bmatrix} R & T \\ 0 & 1 \end{bmatrix} \cdot \begin{bmatrix} x_c \\ y_c \\ z_c \\ 1 \end{bmatrix} \tag{2}$$

*2.2. Hand–Eye Calibration Algorithm Based on ICP*

2.2.1. The ICP Matching Algorithm Is Solved at the Initial Value

The Tsai–Lenz method is employed to obtain the initial values for ICP matching. In the hand–eye calibration model shown in Figure 1, $_{Cam}^{Base}T$ and $_{Cal}^{Tool}T$ are fixed values. $_{Cal}^{Cam}T$ can be obtained through camera calibration, and $_{End}^{Base}T$ can be derived from the robotic arm's forward kinematics equation. The transformation of the camera's coordinate frame relative to the robotic arm's base coordinate frame is represented by Equation (3):

$$_{Cam}^{Base}T = {_{End}^{Base}T} \cdot {_{Tool}^{End}T} \cdot {_{Cam}^{Tool}T} \tag{3}$$

$_{Cal}^{Tool}T$ is a constant, and through transformation, Equation (4) can be derived.

$$_{Cal}^{Tool}T = {_{Base}^{Tool}T} \cdot {_{Cam}^{Base}T} \cdot {_{Cal}^{Cam}T} \tag{4}$$

During the calibration process, the poses of calibration board feature points need to be captured in the camera coordinate frame. Simultaneously, the poses of the robotic arm's end effector coordinate frame must be recorded at different positions. The recorded calibration relationships for the first set are denoted as $_{Cal}^{Tool}T = {_{Base}^{Tool}T_1} \cdot {_{Cam}^{Base}T_1} \cdot {_{Cal}^{Cam}T_1}$, and for the second set as $_{Cal}^{Tool}T = {_{Base}^{Tool}T_2} \cdot {_{Cam}^{Base}T_2} \cdot {_{Cal}^{Cam}T_2}$. A total of $N$ sets are collected ($N \geq 15$), and they are collectively expressed in the form of a homogeneous system of equations, as shown in Equation (5).

$$\begin{bmatrix} _{Cal}^{Tool}T \\ _{Cal}^{Tool}T \\ \vdots \\ _{Cal}^{Tool}T \end{bmatrix} = \begin{bmatrix} _{Base}^{Tool}T_1 \cdot {_{Cam}^{Base}T_1} \cdot {_{Cal}^{Cam}T_1} \\ _{Base}^{Tool}T_2 \cdot {_{Cam}^{Base}T_2} \cdot {_{Cal}^{Cam}T_2} \\ \vdots \\ _{Base}^{Tool}T_n \cdot {_{Cam}^{Base}T_n} \cdot {_{Cal}^{Cam}T_n} \end{bmatrix} \tag{5}$$

By combining the equations from the first and second sets, Equation (6) can be derived:

$$_{Base}^{Tool}T_1 \cdot {_{Cam}^{Base}T_1} \cdot {_{Cal}^{Cam}T_1} = {_{Base}^{Tool}T_2} \cdot {_{Cam}^{Base}T_2} \cdot {_{Cal}^{Cam}T_2} \tag{6}$$

Left-multiplying Equation (6) by $_{Base}^{Tool}T_2^{-1}$ and right-multiplying it by $_{Cal}^{Cam}T_1^{-1}$, where $_{Cam}^{Base}T$ represents the unknowns; through setting $_{Cam}^{Base}T$ as $X$, we obtain Equation (7):

$$_{Tool}^{Base}T_2 \cdot {_{Tool}^{Base}T_1}^{-1} \cdot X = X \cdot {_{Cal}^{Cam}T_2} \cdot {_{Cal}^{Cam}T_1}^{-1} \tag{7}$$

Further simplifying Equation (5) leads to Equation (8):

$$\begin{bmatrix} _{Tool}^{Base}T_2 \cdot {_{Tool}^{Base}T_1}^{-1} \cdot X = X \cdot {_{Cal}^{Cam}T_2} \cdot {_{Cal}^{Cam}T_1}^{-1} \\ _{Tool}^{Base}T_3 \cdot {_{Tool}^{Base}T_2}^{-1} \cdot X = X \cdot {_{Cal}^{Cam}T_3} \cdot {_{Cal}^{Cam}T_2}^{-1} \\ \vdots \\ _{Tool}^{Base}T_n \cdot {_{Tool}^{Base}T_{n-1}}^{-1} \cdot X = X \cdot {_{Cal}^{Cam}T_n} \cdot {_{Cal}^{Cam}T_{n-1}}^{-1} \end{bmatrix} \tag{8}$$

Coupling $^{Base}_{Tool}T_2 \cdot ^{Base}_{Tool}T_1{}^{-1}$ and $^{Cam}_{Cal}T_2 \cdot ^{Cam}_{Cal}T_1{}^{-1}$ results in $A$, $B$ in Equation $AX = XB$, and then integrating Equation (8) with $A$, $B$. Using a two-step approach, we obtain Equation (9):

$$\begin{cases} R_A R_x = R_x R_B \\ R_A T_x + T_A = R_x T_B + T_x \end{cases} \tag{9}$$

Using a two-step approach, the rotation matrix $R$ is first solved, followed by the non-linear calculation of the translation vector $T$. To solve for the rotation matrix $R$, we employ the Rodrigues transformation. This transforms it into the Rodrigues parameter form by utilizing the product of the axis vector $P$ and the rotation angle $\theta$, denoted as $R = \exp(P \times \theta)$. Here, $P$ represents a unit vector, and $\theta$ represents the rotation angle. The rotational vector relationship is calculated using the Skew matrix, as shown in Equation (10):

$$Skew\ (P_A + P_B) \cdot P'_x = P_B - P_A \tag{10}$$

In Equation (10), $P_A$ represents the homogeneous coordinate vector of the feature point in the robotic end effector coordinate frame, $P_B$ represents the homogeneous coordinate vector of the feature point in the camera coordinate frame, and $P'_x$ denotes the inverse hand–eye transformation relationship. Further calculations are performed to derive the rotational vector $P_x$, $P_x = \frac{2P'_x}{\sqrt{1+|P'_x|^2}}$. The solved value of $P_x$ is then substituted into Equation (11) to determine $R_x$.

$$R_x = \left(1 - \frac{|P_x|^2}{2}\right)I + \frac{1}{2}\left(P_x P_x^T + \sqrt{4-|P_x|^2}Skew(P_x)\right) \tag{11}$$

Substitute $R_x$ into Equation (9) to solve for $T_x$.

### 2.2.2. Design of Hand–Eye Calibration Method Based on ICP

Just as a rigid body maintains a consistent transformation relationship in space across various positions, the point cloud feature descriptors of a rigid body exhibits specific coordinate expression relationships at different spatial positions. Point cloud registration involves calculating the coordinate expression relationship between corresponding feature points of the target and source point clouds in space, which allows us to determine the spatial coordinate expression relationship between the two-point clouds. The mathematical model is described as follows: In space, there is a source point cloud $P = \{p_1, p_2, \cdots p_m\}$, and a target point cloud $Q = \{q_1, q_2, \cdots q_n\}$, where $m, n$ represents the number of points in the source and target point clouds, and $m \leq n$. In point cloud registration, both the source and target point clouds are samples of the same object, obtained from different angles or positions. The source point cloud is the one we aim to adjust or transform, so that it aligns as closely as possible with the target point cloud through rigid transformations (translation and rotation). This process is known as point cloud registration, as shown in Equation (12), where $R$ represents the rotation matrix of the rigid transformation, and $T$ is the translation vector.

$$P = R \cdot Q + T \tag{12}$$

Point cloud registration typically consists of two processes: coarse registration and fine registration. Coarse registration aims to establish a preliminary correspondence between two-point clouds. This provides a more accurate initial value for fine registration, reducing computation and enhancing matching efficiency. Currently, the ICP algorithm is one widely used method for fine registration. Due to the relatively low accuracy of traditional Tsai–Lenz hand–eye transformation relationships, there still exist positional discrepancies between the transformed calibration board feature points' point cloud and the points located beneath the robotic arm's tool. ICP calculates the optimal rigid transformation using the least squares method, iteratively solving $R$ and $T$ to minimize the Euclidean distance between corresponding points of the source and target point clouds. This ensures the maximum

possible overlap between the two-point clouds, offering features such as good matching performance and robustness. The steps of hand–eye calibration based on ICP are as follows:

Step (1): Use *D455* camera to capture *m* different poses of the calibration board feature point cloud, referred to as point cloud *P*. Simultaneously, move the robotic arm *TCP* tool to collect *n* sets of point clouds of the calibration board feature points, referred to as point cloud *Q*.

Step (2): Set the hand–eye transformation relationship obtained from Tsai–Lenz as the initial value for ICP point cloud matching. Set the matching error threshold as $\varepsilon_{\min}$ and the maximum number of iterations as $K_{\max}$.

Step (3): Substitute the collected point cloud data into ICP for matching.

Step (4): Iteratively perform matching until the matching convergence conditions are met.

Step (5): Upon convergence, output the rigid transformation relationship, which represents the improved hand–eye calibration relationship in terms of accuracy.

During the ICP matching process, the Euclidean distance $d(i)$ between point cloud *P* and point cloud *Q* is given by Equation (13). In the equation, $p_i, q_i$ represents the corresponding points in point cloud *P* and point cloud *Q*, while *n* represents the maximum number of corresponding point pairs.

$$d(i) = \min_{R,T} \sum_{i=1}^{n} ||Rp_i + T - q_i||_2^2 \tag{13}$$

With each iteration, a new *R* and *T* is generated. The Euclidean distance between the corresponding points in point cloud *P* and point cloud *Q* after the $(k-1)$th iteration's rigid transformation is denoted as $d_{k-1}(i)$. The rotation matrix under this rigid transformation is represented by $R_{k-1}$, and the translation vector is represented by $T_{k-1}$. This is shown in Equation (14).

$$d_{k-1}(i) = \arg\min_{R,T} \left( ||(R_{k-1}p_i + T_{k-1}) - q_i||_2^2 \right) \tag{14}$$

The iteration continues until either the specified maximum iteration count is reached, or the error threshold falls below the predetermined threshold, whichever stopping condition occurs first. Assuming the $(k-1)$th iteration did not converge, the spatial positions of point cloud *P* and point cloud *Q* based on the $k-1$ th iteration's rigid transformation will be continued for the $(k)$th iteration, and the rotation matrix $R_k$, translation vector $T_k$, and Euclidean distance $d_k$ for the kth iteration will be output. This is shown in Equations (15) and (16).

$$d_k = \arg\min_{R,T} \left( \sum_{i=1}^{m_s} ||R'(R_{k-1}p_i + T_{k-1}) + T' - q_i||_2^2 \right) \tag{15}$$

$$R_k = R'R_{k-1}, \ T_k = R'T_{k-1} + T' \tag{16}$$

In the computation process, it is imperative to incorporate an error threshold $\varepsilon_{\min}$ and a specified number of iterations $K_{\max}$ to determine when optimal results should be achieved. Otherwise, the efficiency of point cloud registration will be affected due to the large computational workload. In the point cloud registration process, the point cloud error is $\varepsilon_k = \frac{1}{m_S} \sum_{i=1}^{m_S} ||R_k p_i + T_k - q_i||_2^2$. When the corresponding point cloud error is $\varepsilon_k \leq \varepsilon_{\min}$ for the $(k)$th iteration, the rigid transformation is deemed optimal, and at this point, the corresponding rigid transformation relationship *R* and *T* is outputted. Alternatively, a maximum number of iterations $K_{\max}$ can be set, and when the number of iterations $k \geq K_{\max}$ is reached, the computation is stopped, and the rigid transformation relationship *R* and *T*, at this point, is outputted. This rigid transformation relationship represents the calibrated

relationship between the hand and eye with improved accuracy. The flowchart is shown in Figure 3.

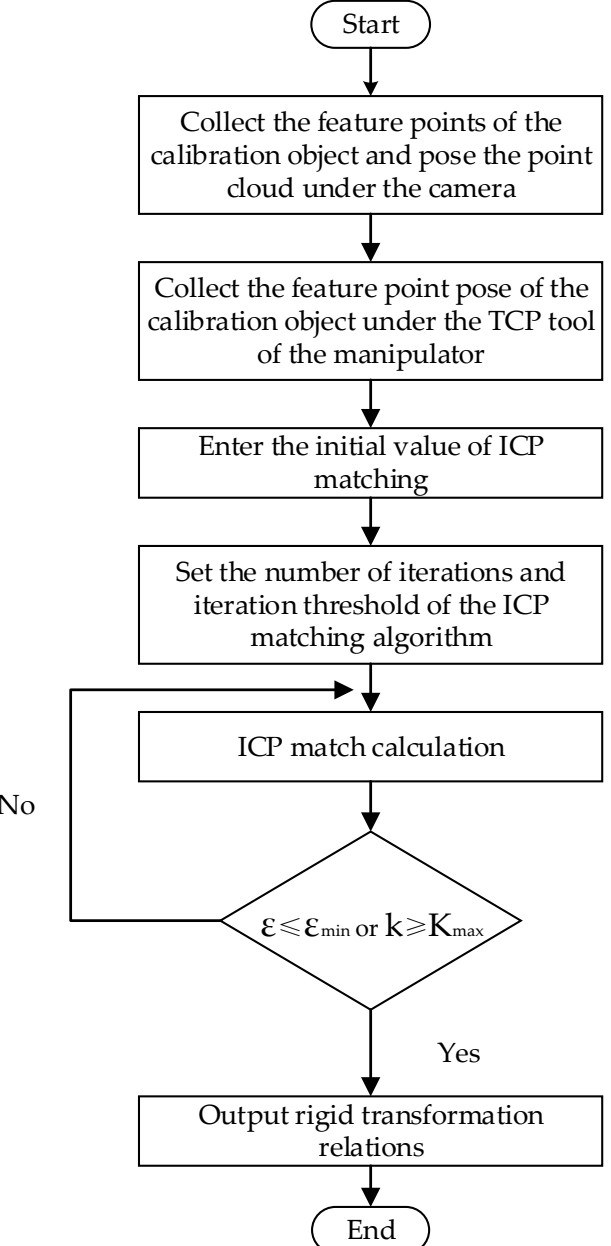

**Figure 3.** Based on the ICP hand–eye calibration method.

## 3. Results and Discussion

To verify the feasibility of the hand–eye calibration method based on ICP, experiments using an apple-picking robot with an eye-in-hand configuration was conducted. The experimental platform, consisting of both hardware and software components, is shown in Figure 4.

The visual sensor used is the RealSense D455 camera, and the UR5e six-axis robotic arm is employed. The specific equipment models and specifications are listed in Table 1. Programming software and simulation platforms such as CloudCompare (v.2.3), Matlab (v.R2021b), and ROS (v.melodic.18.04) were utilized for experimental simulations.

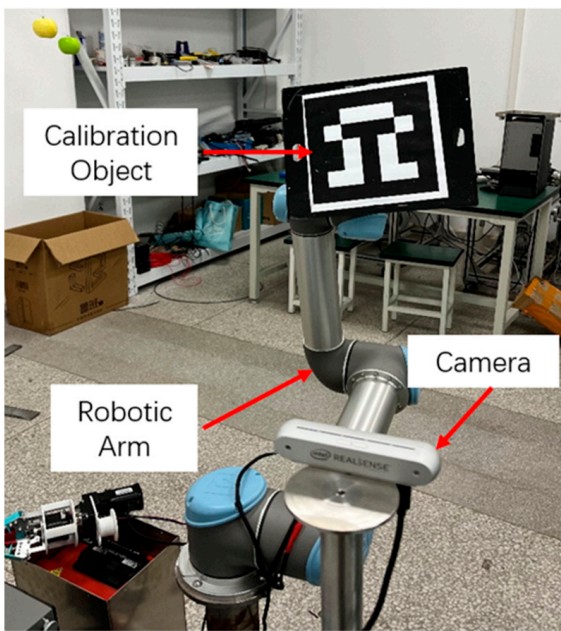

**Figure 4.** Hand–eye calibration experiment site.

**Table 1.** Parameters of experimental instruments and equipment.

| Equipment | Parameter | Name or Numeric Value |
|---|---|---|
| Robotic arm | Model | UR5e |
| | Workspace | $\phi$ 850 mm $\times$ 150 mm |
| | Positioning accuracy | 0.2 mm |
| | Repeatable positioning accuracy | 0.1 mm |
| Vision sensors | Model | Realsense D455 |
| | Measuring range | 60 mm–600 mm |
| | Working distance | 60 mm–150 mm |
| | resolution | 1280 $\times$ 800 |
| Calibration plate | Diameter | 20 mm $\times$ 20 mm |
| | precision | 5 μm |
| PC | CPU | EPC-P3086 |
| | | i9-10900K@3.70GHz |
| | GPU | GPU/Nvidia 1050Ti(12GB) |

To obtain the initial values for ICP iteration in the hand–eye calibration, we gathered the poses of the ArUco-582 calibration board's feature points at various orientations within the camera's field of view. Simultaneously, we recorded the coordinates of the end effector's center point under the robotic arm's base. A total of 20 sets of data were collected. The Tsai–Lenz hand–eye calibration method was used to obtain the initial values for ICP matching, and the calibration process is illustrated in Figure 5.

The resulting rotation matrix and translation vectors are shown in Table 2.

**Table 2.** The rotation matrix and translation vector are obtained based on the Tsai–Lenz hand–eye calibration method.

| Rotation Matrix | | | Translation Vector |
|---|---|---|---|
| −0.258 | 0.014 | 0.965 | −0.499 |
| −0.963 | −0.070 | −0.257 | −0.151 |
| 0.064 | −0.997 | 0.032 | 0.712 |

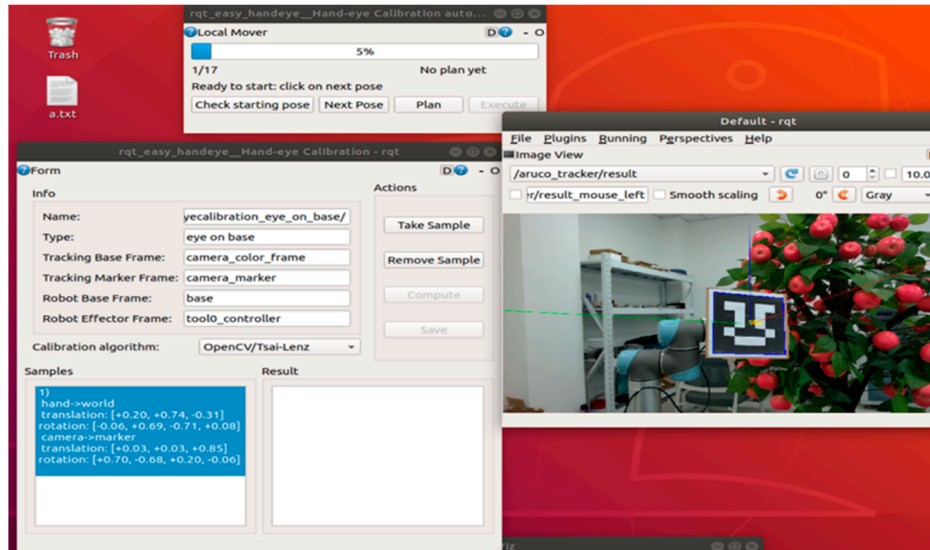

**Figure 5.** The initial value of ICP matching was solved based on the Tsai–Lenz hand–eye calibration method.

To examine the impact of ICP matching initial values and matching parameters on the accuracy of the rigid transformation relationship in the matching output, we utilized the aforementioned solutions as the initial values for the ICP matching process. Comparative experiments were carried out by varying the ICP matching initial values, matching error threshold $\varepsilon_{\min}$, and maximum iteration count $K_{\max}$. The rotational error is computed using Formula (17), while Formula (18) is employed to calculate the translational error:

$$e_R = \frac{1}{n}\sum_{i=1}^{n}||R_A{}^i R_X - R_X R_B{}^i||_F \tag{17}$$

"RARX-RXRB" is a method employed for quantifying errors in hand–eye calibration, commonly utilized to evaluate the disparities between empirically measured values and theoretically estimated values.

RARX: This term denotes the disparity between the actual rotation matrix (R_actual) and the anticipated rotation matrix (R_expected). This matrix encapsulates the rotational association between the camera and the robotic arm's end effector.

RXRB: This term signifies the contrast between the rotation matrix of the robotic arm's end effector (R_arm) and the rotation matrix of the robotic arm's base (R_base). This matrix characterizes the rotational connection of the robotic arm.

Consequently, the "RARX-RXRB" computation quantifies the rotational error between the camera and the robotic arm's end effector, with the influence of the robotic arm's rotation factored out. This calculation imparts valuable insights into the precision of hand–eye calibration by assessing the deviation between the actual and anticipated camera positions, while accounting for the impact of the robotic arm's rotation.

$$e_T = \frac{1}{n}\sum_{i=1}^{n}||R_A{}^i T_X - R_X T_B{}^i - T_x + T_A{}^i||_2 \tag{18}$$

The expression "RXTX-RXTB-TX+TA" constitutes a formula employed for the computation of translational error, involving matrix and vector operations.

RXTX: This term signifies the product of the actual rotation matrix (R_actual) and the actual translational vector (T_actual). It captures the authentic transformational relationship between the camera and the robotic arm's end effector.

RXTB: This term represents the product of the actual rotation matrix (R_actual) and the anticipated translational vector (T_expected). It characterizes the projected transformational relationship between the camera and the robotic arm's end effector.

TX: This refers to the actual translational vector (T_actual).

TA: This symbolizes the projected translational vector (T_expected).

Consequently, the computation expressed as "RXTX-RXTB-TX+TA" shows cases the extent of translational error between the factual and envisaged transformational associations. This calculation serves to evaluate the accuracy of the translational correlation between the camera and the robotic arm's end effector. Lesser error values denote a closer alignment of the actual transformational relationship with the anticipated one, thereby reflecting minimized translational error during the process of hand–eye calibration.

Given that the value of $K_{max}$ remains constant, experiments were conducted to examine the impact of varying ICP matching error thresholds on the accuracy of rigid transformation relationships in the matching outputs, as shown in Figure 6.

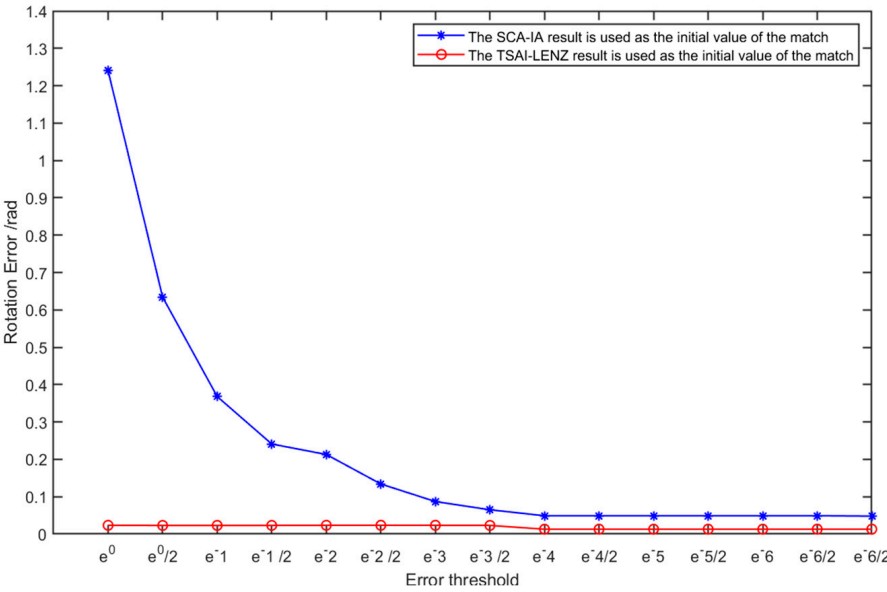

**Figure 6.** Under the same $K_{max}$ premise, the influence of different error thresholds $\varepsilon_{min}$ on the rotational accuracy of the rigid transformation relationship of ICP matching output.

From Figure 6, it can be observed that under the condition of the same maximum iteration count $K_{max}$, the value of the maximum iteration count $K_{max}$ is set to 1000. When the result of Tsai–Lenz hand–eye calibration is used as the initial value for ICP matching, and the matching error threshold $\varepsilon_{min}$ is set to $e^{-4}$, the convergence of the rigid transformation matrix remains unchanged. Similarly, when the rigid transformation relationship from the matching output of the SCA-IA algorithm is employed as the initial value for ICP matching, and the matching error threshold $\varepsilon_{min}$ is set to $e^{-3}$, there is no alteration in the convergence of the rigid transformation matrix. Different initial values for ICP matching lead to variations in the accuracy and efficiency of obtaining the rigid transformation relationship. The accuracy of the rigid transformation relationship obtained when utilizing the result of Tsai–Lenz hand–eye calibration as the initial value for ICP matching is higher than the accuracy achieved using the SCA-IA algorithm.

Similarly, under the condition of the same error threshold $\varepsilon_{min}$, the value of the error threshold $\varepsilon_{min}$ is set to $e^{-7}$. Experiments were conducted to investigate the impact of varying ICP matching maximum iteration counts $K_{max}$ on the accuracy of rigid transformation relationships in the matching outputs, as illustrated in Figure 7.

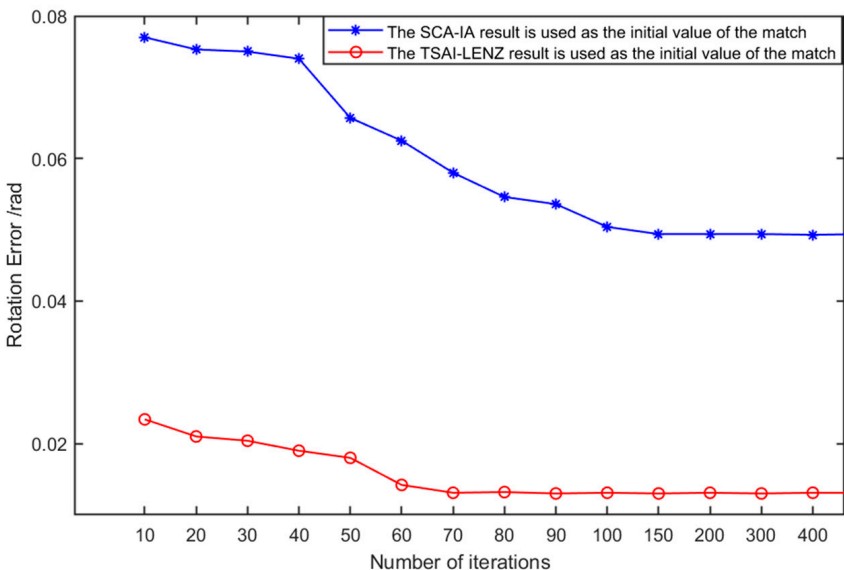

**Figure 7.** Under the same premise of $\varepsilon_{\min}$, the influence of different maximum iterations $K_{\max}$ on the rotational accuracy of the rigid relationship of ICP matching output.

As indicated by the findings presented in Figure 7, when employing the outcomes of the Tsai–Lenz hand–eye calibration as the initial inputs for the ICP matching process, and with a designated matching iteration count denoted by $K_{\max}$ set to 70, the convergence behavior of the rigid transformation matrix remains unaffected. At this juncture, the angular precision error registers at approximately 0.013 radians. Conversely, when utilizing the rigid transformation relationships derived from the output of the SCA-IA algorithm as the foundational values for ICP matching, and with the same designated matching iteration count denoted by $K_{\max}$ set to 150, the rigid transformation matrix successfully converges. Notably, the angular precision error at this convergence point approximates 0.075 radians.

For the purpose of juxtaposing translational errors, a uniform set of ICP matching parameters, denoted by $\varepsilon_{\min}$ and $K_{\max}$, was adopted to facilitate a comprehensive analysis of translational discrepancies. A total of 10 calibration experiments were meticulously conducted. The comparative exhibition of experimental data is meticulously presented in Figure 8.

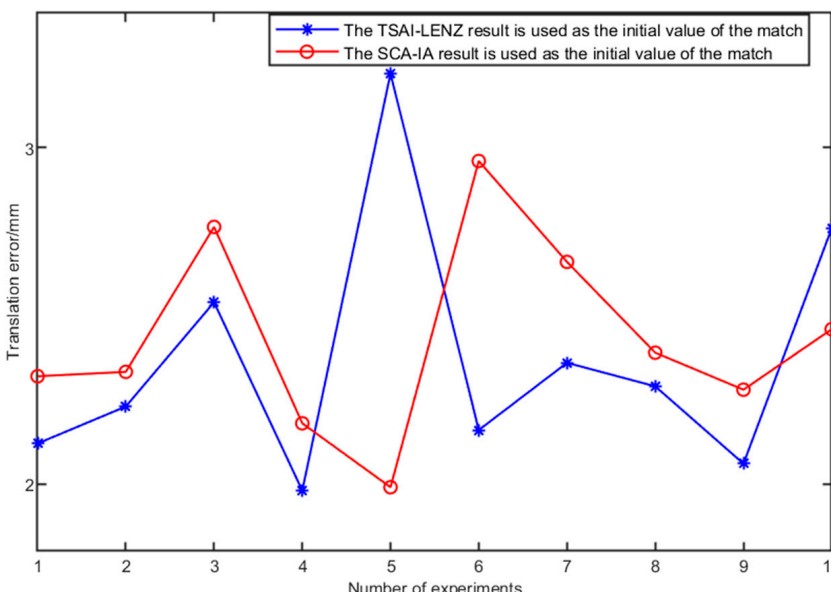

**Figure 8.** The influence of different ICP matching initial values on translation accuracy.

Synthesizing the insights from Figures 7 and 8. It becomes apparent that a higher precision in the initialization of ICP matching leads to swifter convergence. Additionally, it enhances the precision of rotational relationships within the resulting rigid transformations. Notably, Figure 8 reveals that the influence of ICP matching initialization on translational error is relatively modest. This observation emanates from ICP matching's foundation on estimating point cloud centroids, wherein identical point cloud scenarios yield nearly indistinguishable centroid coordinates, thereby yielding minimal disparities in translational error. Incorporating the above-discussed observations leads to the conclusion. This conclusion states that elevated precision in ICP matching initialization enhances matching efficiency. Additionally, it improves the rotational precision of rigid transformation outputs. Consequently, when designating the matching error threshold $\varepsilon_{\min}$ as $e^{-4}$ and setting the maximum iteration count $K_{\max}$ to 100 iterations, an optimal configuration for rigid transformation relationships can be attained.

To further substantiate the efficacy of the ICP-based hand–eye calibration method, a comparative study was conducted through 10 experiments against the traditional dual-quaternion method and Tsai–Lenz method. As shown in Figure 9, it becomes evident that, in comparison to the dual-quaternion method and the Tsai–Lenz method, the ICP-based hand–eye calibration method exhibits superior rotational precision. In terms of translational precision, the ICP-based hand–eye calibration method shows comparable results to the Tsai–Lenz method and outperforms the dual-quaternion method.

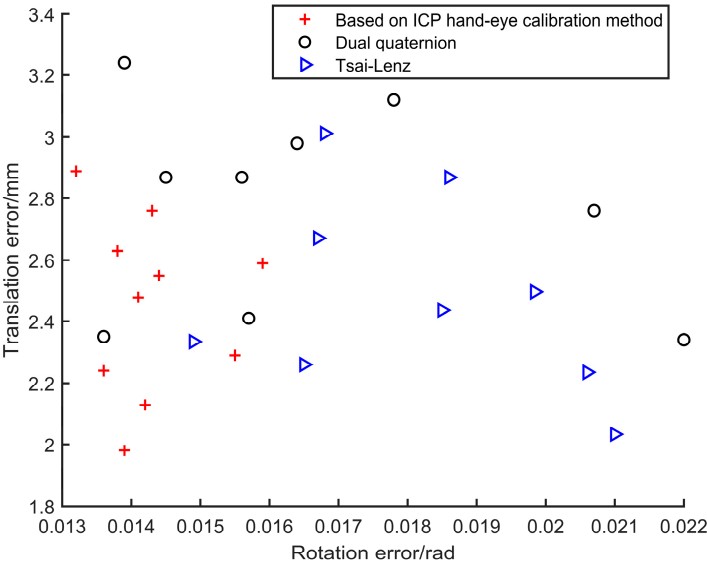

**Figure 9.** Comparison of precision between ICP hand–eye calibration method and other hand–eye calibration methods.

A total of ten comparative experiments were conducted, and the average of the calibration results from these ten trials was computed, as presented in Table 3.

**Table 3.** Hand–eye calibration error.

| Hand–Eye Calibration Method | Rotation Error (rad) | Translation Error (mm) |
| --- | --- | --- |
| Tsai–Lenz | 0.0174 | 2.446 |
| Dual quaternion | 0.0162 | 2.719 |
| Based on ICP hand–eye calibration method | 0.0153 | 2.387 |

From Table 3, it is evident that under the precondition of employing the Tsai–Lenz algorithm's hand–eye calibration results as initial values for the ICP matching, the hand–eye calibration method based on ICP matching demonstrates superior rotational accuracy com-

pared to the Tsai–Lenz algorithm. The rotational precision is enhanced by 13.62%, and the translational precision improves by 2.47%. In comparison to the dual quaternion method, the rotational accuracy witnesses a proportional enhancement of 5.03%, accompanied by a 13.89% increase in translational precision.

## 4. Conclusions

During the traditional hand–eye calibration involving the solution of the $AX = XB$ equation, there has always been precision loss due to decoupling during the linearization process and the accumulation of substantial nonlinear solution errors. Therefore, this paper introduces a hand–eye calibration method based on ICP point cloud matching to enhance calibration precision. Compared with the traditional Tsai–Lenz and dual quaternion methods, the effectiveness of the proposed method is verified by experiments and simulation experiments. For the issue of low initial precision in ICP point cloud matching, which leads to high computational complexity and substantial error in the matched rigid transformation relationship, this study sets the initial values from Tsai–Lenz as the initial values for ICP matching. Additionally, the matching error threshold $\varepsilon_{\min}$ in the matching parameters is set as $e^{-4}$, and the maximum iteration count $K_{\max}$ is set as 70 to optimize both matching efficiency and accuracy. Hand–eye calibration experiments are conducted using a UR5e robotic arm. The results demonstrate that the precision of rotational components can be improved by 13.6%, and translational precision can be enhanced by 2.47% compared to the Tsai–Lenz algorithm. When compared to the dual quaternion method, the rotational precision enhancement is 5.03%, while translational precision improves by 13.89%. This method holds certain significance for enhancing operational efficiency in the context of apple-picking robots.

**Author Contributions:** Conceptualization, T.Y. and P.L.; methodology, T.Y.; software, T.Y.; validation, P.L., Y.L. and T.J.; formal analysis, T.Y.; investigation, H.Y.; resources, G.C.; data curation, T.Y.; writing—original draft preparation, T.Y.; writing—review and editing, P.L.; visualization, T.J.; supervision, Y.L. All authors have read and agreed to the published version of the manuscript.

**Funding:** This research was funded by the National Natural Science Foundation of China (61903184); Natural Science Foundation of Jiangsu Province Youth Fund (BK20181017, BK2019K186); 2021 Provincial Key R&D Plan (Industry Foresight and Common Key Technologies) (BE2021016-5).

**Institutional Review Board Statement:** Not applicable.

**Data Availability Statement:** Not applicable.

**Acknowledgments:** The authors gratefully acknowledge the editors and anonymous reviewers for their constructive comments on our manuscript.

**Conflicts of Interest:** The authors declare no conflict of interest.

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
