# Peer review of "Research on Hand–Eye Calibration Accuracy Improvement Method Based on Iterative Closest Point Algorithm"

_agriculture, doi:10.3390/agriculture13102026_

Round 1

Reviewer 1 Report

1. Picture 1 and 5 needs to be enlarged to make the contents in the pictures clearer.

2. [Line 79-80] The author needs to explain the difference between the two camera mounting methods: eye-in-hand and eye-to-hand.

3. [Line 86] What does "ggipper" mean.

4.[Line225-226] The sentence needs to be rewritten to make it easier to understand. (What does "rigid rigid transformation relationship" mean?)

5. The annotations in picture 4 need to be modified to make the picture easier to understand.

6. How is the camera in picture 4 attached to the arm and can it move with the arm? The position of the camera in the picture does not seem to allow the calibration object to be captured.

6. In Section 3.1.3, the test is conducted in a messy scene. Does this generate a lot of noise?

7. [Line 315] This sentence is incomplete.

Author Response

Response to reviewers

We gratefully appreciate for your valuable suggestion

1.Comment:

Picture 1 and 5 needs to be enlarged to make the contents in the pictures clearer.

1.Reply:

Thank you for your feedback. We have taken your suggestion into account and have already enlarged Picture 1 and Picture 5 to improve the clarity of the content. Please find the enlarged versions of the images below

Picture1 Hand-eye calibration model.

Picture5: The initial value of ICP matching was solved based on the Tsai-lenz hand-eye calibration method.

2.Comment:

[Line 79-80] The author needs to explain the difference between the two camera mounting methods: eye-in-hand and eye-to-hand.

2.Reply:

Eye on Hand: In this scenario, the camera (eye) is mounted on the end effector of the robot arm. This means the camera moves along with the robot's end effector. The calibration goal in this case is to determine how the camera's coordinate system corresponds to the robot arm's coordinate system.

Eye Outside Hand: In this scenario, the camera is fixed in position and doesn't move with the robot arm's motion. The camera might be located outside the robot's workspace but is able to view the robot arm and the operating area. The calibration goal here is to establish the coordinate transformation relationship between the camera's coordinate system and the robot arm's coordinate system, while considering the spatial relationship between the camera and the robot's base.

3.Comment:

[Line 86] What does "ggipper" mean.

3.Reply:

Thank you very much for carefully reviewing my paper and bringing to my attention the misspelling of "ggipper." I apologize for this oversight. The correct spelling should be "gripper." I will make the necessary corrections in the relevant sections of the paper to ensure accuracy and consistency.

4.Comment:

[Line225-226] The sentence needs to be rewritten to make it easier to understand. (What does "rigid rigid transformation relationship" mean?)

4.Reply:

I would like to express my gratitude for your meticulous review of my work and for bringing to light the error in the phrase "rigid rigid transformation relationship." I apologize for this oversight in the text. The accurate phrase should be "rigid transformation relationship." I sincerely appreciate your attention to detail in identifying this mistake.

5.Comment:

The annotations in picture 4 need to be modified to make the picture easier to understand.

5.Reply:

The annotations might have been simplified. I have re-annotated them.

6.Comment:

How is the camera in picture 4 attached to the arm and can it move with the arm? The position of the camera in the picture does not seem to allow the calibration object to be captured. 6. In Section 3.1.3, the test is conducted in a messy scene. Does this generate a lot of noise?

6.Reply:

(1)In Figure 4, because the approach used is "eyes on the outside of the hand," the camera is placed on the floor along with the robotic arm. Please refer to this image for reference.

(2)By employing an "eye outside the hand" approach, the calibration object is positioned at the end of the robotic arm for capture by the camera.

(3)During the experimental process, noise is inevitably generated, with the primary source of noise originating from camera data acquisition.

7.Comment:

[Line 315] This sentence is incomplete.

7.Reply:

I have refined the sentence.

.

Figure 9. Comparison of Precision Between ICP Hand-Eye Calibration Method and Other Hand-Eye Calibration Methods

Reviewer 2 Report

Using the solution result of Tsai-Lenz as the initial value of the ICP matching algorithm, a hand-eye calibration method based on ICP algorithm is proposed to to improve the calibration accuracy. The method is verified by experimental simulation compared with the traditional Tsai-Lenz and dual quaternion methods. The paper has  innovation and practicality. But the following issues in the paper need to be further elaborated and explained.

1. Please briefly explain the meaning of the symbols in the equation of line 36.

2. Please verify the correctness of Equation (1) and how to eliminate zc in the calculation process .

3. Figure 5 cannot represent the calibration process clearly,and it is more appropriate to represent Table 2 with rotation matrix R and a translation matrix T

4.  The values of  Kmax for Figure 6  and   for  Figure 7 should  be  provided.

5. Why is the TSAI-LEN result is larger than The SCA-IA result when the number of experiments is 5.

6. In each legend of Figure 9, the number of experiments can be added as a marker to make the comparison effect more pronounced.

7.Equation (12) should be Equation (13) in line 200

1. There are problems with the English expression of some professional terms in the paper, and improvement is needed.Such as

AGV automated guided vehicles (in Line 28 to Line29),fractured rigid body fracture surface(line 62),under the camera frame(In line 72), ggipper (in line86)

2.There are spelling errors in the paper, which need to be carefully read and corrected

Author Response

Response to reviewers

We feel great thanks for your professional review work on our article. As you are concerned, there are several problems that need to be addressed. According to your nice suggestions, we have made extensive corrections to our previous draft, the detailed corrections are listed below.

1.Comment:

Please briefly explain the meaning of the symbols in the equation of line 36.

1.Reply:

In hand-eye calibration, the equation AX=XB signifies the transformation relationship between the camera coordinate system and the robotic arm coordinate system. Here, A and B represent the transformation matrices of the camera and the robotic arm, respectively. X represents the solution sought in hand-eye calibration, denoting the transformation relationship between the camera and the robotic arm. Essentially, this equation describes the translation and rotation relationship between the camera and robotic arm coordinate systems. By solving this equation, accurate results of hand-eye calibration can be obtained, enabling precise spatial positioning and control.

2.Comment:

Please verify the correctness of Equation (1) and how to eliminatein the calculation process

2.Reply:

I apologize for the error in the notation of Formula (1). It has been corrected now.

3.Comment:

Figure 5 cannot represent the calibration process clearly,and it is more appropriate to represent Table 2 with rotation matrix R and a translation matrix T

3.Reply:

To obtain initial values for ICP matching, the Tsai-Lenz hand-eye calibration method is employed in this study. This process is based on ROS Melodic for acquiring initial values for ICP matching. By utilizing relevant hand-eye calibration packages within ROS, such as the UR robotic arm driver package, Realsense package, and Aruco calibration board package, initial values for ICP matching are obtained. A minimum of 17 data sets need to be collected to complete the calibration process.

4.Comment:

4.The values of  for Figure 6 and for Figure 7 should be provided

4.Reply:
Under the same maximum iteration count, the maximum iteration count  is set to 1000.

With the same error threshold, the error threshold  is set to .

5.Comment:

Why is the TSAI-LEN result is larger than The SCA-IA result when the number of experiments is 5.

5.Reply:

It could be due to experimental-induced human error or the influence of noise during the camera data acquisition process.

6.Comment:

In each legend of Figure 9, the number of experiments can be added as a marker to make the comparison effect more pronounced

6.Reply:

Considering how to succinctly and intuitively understand the translation and rotation errors of each method in the images, the number of experiments conducted has not been included.

7.Comment:

Equation (12) should be Equation (13) in line 200

7.Reply:

I have noticed an error in the labeling of a formula in the paper, and I sincerely apologize for this. I have made the necessary correction.

Comments on the Quality of English Language

1.There are problems with the English expression of some professional terms in the paper, and improvement is needed.Such as

AGV automated guided vehicles (in Line 28 to Line29),fractured rigid body fracture surface(line 62),under the camera frame(In line 72), ggipper (in line86)

2.There are spelling errors in the paper, which need to be carefully read and corrected

Thank you for your valuable feedback. We appreciate your diligence in reviewing our paper. We apologize for any spelling errors that may have been present. We will thoroughly review the paper again to identify and correct these mistakes. Your suggestions will undoubtedly contribute to the overall quality of the paper. If you have any further comments or recommendations, please feel free to share them. We are committed to ensuring the accuracy and clarity of the paper. Thank you for your time and assistance

Reviewer 3 Report

Dear Authors,

In your paper, you are presenting an improved method to calibrate hand-eye cameras based on the ICP algorithm. Overall, your paper is not well written and not clear, making it more difficult to assess its soundness. The topic is borderline compared to the journal topic. The overall contribution to the calibration seems minimal.

Regardless of this point, in my opinion, this work brings no clear contribution to the general topic of camera calibration and the fact that it is poorly written doesn't help at all.

I can list some comments too:

  • The proposed method is compared only to Tsai-Lenz and the quaternion method. What about other common approaches?
  • What's new in your method compared to others using ICP as well?
  • What happens when this method is applied to an actual agricultural environment? As an example, how this method behaves with the typical lighting issue that is always present in outdoor operations?
  • Most of the symbols used are introduced but never explained if not later. Several time the same symbols is used for different quantities
  • Quality of fig.1 is very poor
  • the way symbols and equations are written is odd, sometimes they float compared to the rest of the text, a rectangle appears in place of the product symbol,
  • the x-axis of figure 6 is poorly formatted and difficult to understand

Thus, in my opinion, this paper cannot be accepted.

The quality of the English language is poor.

There are many typos, but even worse, many sentences are not clear and difficult to interpret.

Author Response

Response to reviewers

We gratefully appreciate for your valuable suggestion

1.Comment:

in your paper, you are presenting an improved method to calibrate hand-eye cameras based on the ICP algorithm. Overall, your paper is not well written and not clear, making it more difficult to assess its soundness. The topic is borderline compared to the journal topic. The overall contribution to the calibration seems minimal.

  1. Reply:

1: The method proposed in this paper is an ICP-based hand-eye calibration approach. The ICP algorithm requires an initial value for matching, and in this study, we utilize the outcomes of the traditional Tsai-Lenz hand-eye calibration method as the initial values for the ICP matching process. This strategy aims to enhance the precision of the calibration process.

2: Addressing the concern about the alignment of our paper with the journal's scope, our research is intricately linked to the broader context of apple-picking robotics. Through the hand-eye calibration, we determine the relative positions of the camera and robotic arm. This facilitates the robot in efficiently gripping apples based on their visual identification.

3: Apologies for any shortcomings in the writing quality. I commit to meticulously reviewing and revising the manuscript to ensure its clarity and coherence.

2.Comment:

The proposed method is compared only to Tsai-Lenz and the quaternion method. What about other common approaches?

2.Reply:

(1)The method we propose utilizes the results of the Tsai-Lenz hand-eye calibration method as initial values for ICP matching, and these values are compared with the initial values obtained from the SCA-IA method. The results indicate that using the results of the Tsai-Lenz hand-eye calibration method as ICP initial values yields better outcomes than the initial values obtained from the SCA-IA method.

(2): We used the outcomes of the conventional Tsai-Lenz hand-eye calibration method as initial values for the ICP matching process. Additionally, we compared these results with the outcomes obtained using the Tsai-Lenz hand-eye calibration method itself. The findings indicated improvements in both rotational and translational precision. This suggests that employing the results of the Tsai-Lenz hand-eye calibration method as initial values for ICP matching enhances the accuracy of the hand-eye calibration relationship. This confirms the effectiveness of our method. Moreover, in comparison with the dual quaternion method, the results also demonstrated slight enhancements in both translational and rotational accuracy.

3.Comment:

What's new in your method compared to others using ICP as well?

3.Reply:

In contrast to other ICP algorithms that obtain initial values using methods like SAC-IA (Sample Consensus Initial Alignment) and PCA (Principal Component Analysis), which also affect the accuracy of ICP matching, our approach directly utilizes the outcomes of the Tsai-Lenz hand-eye calibration for ICP matching initial values. This choice mitigates the influence of lower precision in initial values on the accuracy of ICP matching.

4.Comment:

What happens when this method is applied to an actual agricultural environment? As an example, how this method behaves with the typical lighting issue that is always present in outdoor operations?

4.Reply:

The outcomes of outdoor hand-eye calibration have been less than optimal, possibly due to the influence of lighting conditions on the camera and potential inaccuracies in data collection. To address this, one possible solution is to perform the hand-eye calibration indoors, where lighting can be controlled, and subsequently deploy the robot for outdoor harvesting tasks. Another option could involve implementing some form of shading for the camera to mitigate the impact of varying lighting conditions.

5.Comment:

Most of the symbols used are introduced but never explained if not later. Several time the same symbols is used for different quantities

5.Reply:

Regarding the many symbols you've pointed out without explanations, I will carefully revise the document and provide explanations for them in the subsequent version.

6.Comment:

:the way symbols and equations are written is odd, sometimes they float compared to the rest of the text, a rectangle appears in place of the product symbol,

6.Reply:

The Word version is fine, but in the PDF version, there's an issue with rectangular multiplication symbols appearing. I will address this in the subsequent revisions.

7.Comment:

the x-axis of figure 6 is poorly formatted and difficult to understand

7.Reply:

The formatting issue on the x-axis has been corrected due to the small size of the images.

Reviewer 4 Report

1.    It is suggested to revise the paper, including formatting, charts, formulas, etc. 

2.    In the introduction, many expressions need to be modified and supplemented, such as letters that appear for the first time and comments that are required, such as line36, 52, 46, etc.

3.    In Materials and Methods, It is suggested to introduce the method and principle of hand-eye correction in more detail. Also check the expression on lines 82-85.

4.    In table 3, how are the errors measured by several methods of hand-eye calibration calculated? It is suggested to express in detail how the standardized true values are measured.

Author Response

Response to reviewers

We gratefully appreciate for your valuable suggestion

1.Comment:

It is suggested to revise the paper, including formatting, charts, formulas, etc.

1.Reply:

Thank you for your valuable feedback. We greatly appreciate your suggestions for revising the paper, including aspects such as formatting, charts, and formulas. We acknowledge the importance of ensuring the quality and clarity of the paper, and we are committed to making the necessary improvements to address the mentioned issues. We are dedicated to refining the paper to meet the highest standards. Once again, we thank you for your time and assistance in improving our paper.

2.Comment:

In the introduction, many expressions need to be modified and supplemented, such as letters that appear for the first time and comments that are required, such as line36, 52, 46, etc.

2.Reply:

Thank you for your feedback. We appreciate your observation regarding the introduction section. We will carefully review and revise the expressions, especially when introducing new terms and providing necessary comments.

3.Comment:

In Materials and Methods, It is suggested to introduce the method and principle of hand-eye correction in more detail. Also check the expression on lines 82-85.

3.Reply:

Thank you for your suggestion. In the Materials and Methods section, we will provide a more comprehensive explanation of the method and principles of hand-eye calibration. We will offer additional details to ensure readers have a clear understanding of the steps and underlying principles of the hand-eye calibration method we employed.

4.Comment:

In table 3, how are the errors measured by several methods of hand-eye calibration calculated? It is suggested to express in detail how the standardized true values are measured.

4.Reply:

Thank you for bringing this to my attention. I appreciate your feedback. I will definitely make sure to include the error calculation in the paper as per your suggestion.

The error is calculated using the following formula:

                                                              (1)

"RARX-RXRB" is a method used for calculating errors in hand-eye calibration, commonly employed to assess the discrepancies between actual measured values and theoretical estimated values.

RARX: This refers to the difference between the actual rotation matrix (R_actual) and the expected rotation matrix (R_expected). This matrix represents the rotational relationship between the camera and the robotic arm's end-effector.

RXRB: This refers to the difference between the rotation matrix of the robotic arm's end-effector (R_arm) and the rotation matrix of the robotic arm's base (R_base). This matrix represents the rotational relationship of the robotic arm.

Therefore, the "RARX-RXRB" calculation signifies the rotation error of the camera and the robotic arm's end-effector, with the influence of the robotic arm's rotation removed. This calculation provides insights into the accuracy of the hand-eye calibration by evaluating the difference between the actual and expected camera positions while considering the effects of the robotic arm's rota

                                                 (2)

"RXTX-RXTB-TX+TA" signifies a formula employed to compute translational error, involving matrix and vector operations.

RXTX: This term represents the product of the actual rotation matrix (R_actual) and the actual translational vector (T_actual). It captures the real transformation relationship between the camera and the robotic arm's end-effector.

RXTB: This term stands for the product of the actual rotation matrix (R_actual) and the expected translational vector (T_expected). It denotes the anticipated transformation relationship between the camera and the robotic arm's end-effector.

TX: This refers to the actual translational vector (T_actual).

TA: This signifies the expected translational vector (T_expected).

Consequently, the computation of "RXTX-RXTB-TX+TA" illustrates the translational error between the actual and anticipated transformation relationships. This calculation facilitates the evaluation of the precision of the translational relationship between the camera and the robotic arm's end-effector. Smaller error values denote a closer alignment of the actual transformation relationship with the expected one, thereby reflecting reduced translational error during hand-eye calibration.

Round 2

Reviewer 3 Report

Dear Authors,

Your paper has been slightly improved, but, to me, it is still lacking and needs to be extensively revised. English language is not always clear making some of your reasoning difficult to understand. Moreover, the results in some of your figures contradict what you have written. But in general, the comments on your results are very limited. Also, you could have added the salient parts of your replies to my comments in the text. They have little to no value if they stay in the cover letter.

Here I will list my comments following the order of your paper, so minor and major comments are mixed together:

1. Lines 30-31: "[the target] position and pose information, which is converted into a robotic arm". This means that the target pose becomes a robotic arm

2. Fix the punctuation. Often there are more dots or commas than needed. Also, the spacing after punctuation should be fixed

3. Lines 45-46. "Common linear or nonlinear [...]", this sentence has no verb

4. The name of the first author of a reference is often written in capital letters, but sometimes only the first letter is capitalised. Use the same format for both according to the journal guidelines.

5. Line 51. "Solve R, T [...]". No explanation of what R and T are. I can guess what they are, but why should I rely on guessing? Also, the symbol T is used both for representing translations and homogeneous transformation. Try to avoid ambiguous symbols (this was already commented on in the previous round): in this case, for example, you can follow the convention by using a lowercase t for the vector and a capitalised T for the matrix.

6. Acronyms should be explicitly written at least the first time they appear. e.g., Singular Value Decomposition (SVD) on line 55 or ICP in the abstract.

7. line 55. why "at the same time" is between two commas

8. Line 56 "Qi et al. [19] solve [...]" sentence is incomplete. Say what they are solving

9. Why is there a section 2.1 if there is no section 2.2?

10. Line 92. You state "[The vision system] does not lose target information with the movement of the robotic arm". Yet if the robotic arm is between the target and the camera this is not true.

11. Never justified why the authors chose an eye-to-hand architecture instead of an eye-in hand too. Given the underlying agriculture scenario where the robot arm interacts with an apple or crops in general, isn't an eye-in-hand architecture superior since it is less affected by obstruction from the robot itself but also from the foliage or the environment in general?

12. Line 94. The sentence is split in two by fig.1

13. Line 98-99. "Object" is misspelled

14. Since you are referring a lot to the eq. AX = BX, maybe you should properly label it as an equation the first time and then refer back to it when needed.

15. Lines 103-105. The sentence is not very clear based on how it is written (what is the principle of a point?) and some extra but key pieces of information are missing. How does the camera detect the apple pose? How does a pose (= position + orientation) become a single point? what does the single point represent? Does the apple orientation information get lost?

16. Equation 1. Why z_C is multiplying [u v 1]^T?

17. The multiplication symbol issue is still there

18. Line 125. Why is there a section 3.1 if there is no section 3.2?

19. Most of the content of section 3 apart from the actual results and the relative discussion should be part of "the material and methods" section.

20. Line 129. What does "positive kinematics" mean? Direct kinematics?

21. Line 133. ToolCalT is fixed but never explained why (the calibration target is held by the robot)

22. Line 133 again. Not clear how a fixed ToolCalT transforms eq3 into eq4.

23. Line 137. The sentence is poorly written.

24. Line 144. The sentence is very unclear and the fact that there are mistakes in eq7 (see next point) makes it even worse.

25. Eq7, left-hand side of all equations. It should be BaseToolTn-1 BaseToolTn-1X

26. Line 147 and eq8. I see no difference between eq7 and eq8.

27. Line 150 "integrating" may be a misleading term.

28. Eq.9 Various R and T symbols were never introduced.

29. Line 155. There is no rotation matrix A. There is the transformation matrix A or the rotation matrix RA

30. Lines 157-158. P is poorly defined (unit vector of what?).

31. Line 158. "Skew" is not a function. What you are doing is computing the skew matrix of PA+PB.

32. Eq10. What does the prime ' mean for you? Transpose? Using T is more common in my opinion, but you can keep ' if you explain it.

33. Line 163. What is the "backhand-eye transformation relationship"

34. Line 164. "Further calculate the rotation vector Px" and "the solve px substitution equation 11 solves for Rx". Write proper sentences

35. Line 167. There are no A or B in eq9.

36. Line 169 (and in other places too). You are using "definite" instead of "defined".

37. Line 193 and in other parts. What is a "rigidity transformation"?

38. Line 213-214. What is the "maximum number of corresponding point logarithms"? Is it the same n as defined in line 176? If so, what happens to eq13 when m < i <= n ?

39. Lines 216-218. Very poorly written.

40. Eq15. What is ms? and why later did it become MS? Are they the same thing?

41. You often refer to [something] in a reference frame as "[something] under the reference frame". It seems an odd way to phrase it to me. Using "in the reference frame" or "with respect to (w.r.t.) the reference frame" seems the more common way to convey this concept

42. line 249. "The vision sensor uses a [camera] and a [robotic arm]". This sentence makes no sense. The robot is used by the vision sensor, and the camera IS the vision sensor.

43. Lines 250-252. "Applied to programming [...]". The sentence makes no sense. Also, what are the other experimental simulation platforms?

44. The authors should discuss the experimental setup more. What is the effect of the relative pose of the camera with respect to the robot (e.g., what happens when the camera is quite far from the robot? or what happens when it is very close)? What about the lighting (outdoor applications suffer a lot from inconsistent lighting but the problem is never acknowledged)? 

45. Fig5 caption is practically unrelated to the content of fig5.

46. Fig.6 The error using your method (blue line) is always worse (larger)

47. Discussion on Fig.6 is unclear

48. Why are you using powers of e as thresholds? Why not use a logarithmic scale at this point?

49. Fig7. Again your method has always a larger error

50. It seems from the comments on fig7 that the curves are labelled wrong (or the comments refer to the wrong curve). So what are the actual data? I imagine the same confusion holds for all your plots, so it is impossible for me to give some feedback...

51. Fig8 and its comments. You state that the initial value of ICP has little effect. It has never been explained how (and how much )the ICP changed between the 10 experiments. So it is difficult to assess if your assumption is general or true only for your limited set of data.

52. Line 324. "This is compared to [...]" What is "this"?

53. All parts related to eqs.17-18 should be moved to the methods section or at least before figure 9 since the figure is obtained by the two equations.

54. Lines 366-367. What does "year-on-year" mean

55. You are not reporting the limitations of your method. Are there no limitations? Your results are general and always true?

The quality of English language is still low. I pointed out some issues in my list, but I suggest the authors to carefully revise the paper.

Author Response

We feel great thanks for your professional review work on our article. As you are concerned, there are several problems that need to be addressed. According to your nice suggestions, we have made extensive corrections to our previous draft, the detailed corrections are listed below.

  1. Lines 30-31: "[the target] position and pose information, which is converted into a robotic arm". This means that the target pose becomes a robotic arm

1.Reply:the 3D vision camera senses the target and obtains its position and pose information, which is converted 31 into a robotic arm through computer processing to execute grasping instructions

  1. Fix the punctuation. Often there are more dots or commas than needed. Also, the spacing after punctuation should be fixed

2.reply:

Thank you very much for your feedback and guidance. Regarding the issue you mentioned, we will make the necessary modifications and adjustments.

  1. Lines 45-46. "Common linear or nonlinear [...]", this sentence has no verb

3.reply: The sentence has been modified.

  1. The name of the first author of a reference is often written in capital letters, but sometimes only the first letter is capitalised. Use the same format for both according to the journal guidelines.

4.reply: The formatting has been adjusted.

  1. Line 51. "Solve R, T [...]". No explanation of what R and T are. I can guess what they are, but why should I rely on guessing? Also, the symbol T is used both for representing translations and homogeneous transformation. Try to avoid ambiguous symbols (this was already commented on in the previous round): in this case, for example, you can follow the convention by using a lowercase t for the vector and a capitalised T for the matrix.

5.reply: In various chapters and sections, both R and T are used to represent rotation matrices and translation vectors. The use of uppercase R and T is intended to provide a clearer distinction for rotations and translations.

  1. Acronyms should be explicitly written at least the first time they appear. e.g., Singular Value Decomposition (SVD) on line 55 or ICP in the abstract.

6.reply: The sentence has been modified.

  1. line 55. why "at the same time" is between two commas

7.reply: Due to a writing error,The sentence has been modified.In comparison to the two-step approach, this method constructs a new set of linear equations using quaternions and then employs SVD(Singular Value Decomposition) for the simultaneous solution of .

  1. Line 56 "Qi et al. [19] solve [...]" sentence is incomplete. Say what they are solving

8.reply: QI addresses the issue of coupled precision loss in the solution process and proposes a method that combines the characteristics of data on SO(4) based on the dual quaternion algorithm to achieve the solution.

  1. Why is there a section 2.1 if there is no section 2.2?

Reply: Thank you very much for your feedback. I also recognize this issue. The formatting of the sections has been adjusted.

  1. Line 92. You state "[The vision system] does not lose target information with the movement of the robotic arm". Yet if the robotic arm is between the target and the camera this is not true.

10.reply: Based on what you've mentioned, you're considering the scenario where the robotic arm obstructs the camera during its movement, resulting in the camera being unable to detect the target. However, in this situation, regardless of how the robotic arm moves, the target's position remains constant relative to the camera. To address partial obstruction by the robotic arm, adjustments can be made to both the camera's position and the robotic arm's orientation.

  1. Never justified why the authors chose an eye-to-hand architecture instead of an eye-in hand too. Given the underlying agriculture scenario where the robot arm interacts with an apple or crops in general, isn't an eye-in-hand architecture superior since it is less affected by obstruction from the robot itself but also from the foliage or the environment in general?

11.reply: In the "eye-in-hand" scenario, the camera is mounted on the robotic arm's end effector and moves along with the robot's motion. In the "eye-to-hand" scenario, the camera remains fixed in position but can observe the robotic arm and the operating area. The "eye-to-hand" visual system maintains a fixed field of view, ensuring that target information is not lost as the robotic arm moves. The calibration objective in this scenario is to establish the transformation relationship between the camera frame and the robotic arm frame.

  1. Line 94. The sentence is split in two by fig.1

12.reply: Format modified。

  1. Line 98-99. "Object" is misspelled

13.reply: modified

  1. Since you are referring a lot to the eq. AX = BX, maybe you should properly label it as an equation the first time and then refer back to it when needed.

14: For intuitive reading and comprehension

  1. Lines 103-105. The sentence is not very clear based on how it is written (what is the principle of a point?) and some extra but key pieces of information are missing. How does the camera detect the apple pose? How does a pose (= position + orientation) become a single point? what does the single point represent? Does the apple orientation information get lost?

Reply: In this case, Figure 2 purely illustrates the relationships between the pixel coordinate frame, image coordinate frame, and camera coordinate frame, as well as the method for representing a point in each of these frames. Certainly, a 3D camera can acquire the pose of the apple through algorithms. Currently, in this project, the apple's pose is mainly obtained by determining the straight line formed by connecting two points on the apple, and representing this line with a vector, thereby obtaining the apple's pose information. The focus of this study is hand-eye calibration. In this study, hand-eye calibration is based on ROS (Robot Operating System). During the hand-eye calibration process on the ROS platform, the calibration board (ARUCO-582) will display its pose in the camera coordinate frame.

  1. Equation 1. Why z_C is multiplying [u v 1]^T?

Reply:

For the sake of conciseness, the specific process is as follows:

(1)

(2)

    (3)

(4)

(5)

  1. The multiplication symbol issue is still there

Reply: The formula has been modified.

  1. Line 125. Why is there a section 3.1 if there is no section 3.2?

Reply: Thank you very much for your feedback. I also recognize this issue. The formatting of the sections has been adjusted.

  1. Most of the content of section 3 apart from the actual results and the relative discussion should be part of "the material and methods" section.
  2. Line 129. What does "positive kinematics" mean? Direct kinematics?

Reply: My apologies for the confusion.Should be "forward kinematics."

  1. Line 133. ToolCalT is fixed but never explained why (the calibration target is held by the robot)

Reply: The calibration board is fixed at the end position of the robotic arm. This is determined by the hand-eye calibration method with an extrinsic camera setup.

  1. Line 133 again. Not clear how a fixed ToolCalT transforms eq3 into eq4.

Reply: This pertains to robot kinematics, representing a transformation from one coordinate system to another.

  1. Line 137. The sentence is poorly written.

Reply: The sentence has been modified.

  1. Line 144. The sentence is very unclear and the fact that there are mistakes in eq7 (see next point) makes it even worse.
  2. Eq7, left-hand side of all equations. It should be BaseToolTn-1BaseToolTn-1X

Reply: Thank you for your careful review. The sentence and formulas have been revised.

  1. Line 147 and eq8. I see no difference between eq7 and eq8.

Reply: Thank you for your careful review. The sentence and formulas have been revised.

  1. Line 150 "integrating" may be a misleading term.
  2. Eq.9 Various R and T symbols were never introduced.
  3. Line 155. There is no rotation matrix A. There is the transformation matrix A or the rotation matrix RA
  4. Lines 157-158. P is poorly defined (unit vector of what?).
  5. Line 158. "Skew" is not a function. What you are doing is computing the skew matrix of PA+PB.
  6. Eq10. What does the prime ' mean for you? Transpose? Using T is more common in my opinion, but you can keep ' if you explain it.

Reply:Regarding questions 27 to 32, specific content and formatting can be referred to in the paper '[16] Tsai, R.Y.; Lenz, R.K. A new technique for fully autonomous and efficient 3D robotics hand/eye calibration. IEEE Trans Robot Autom 1989, 5, 345–358.' This paper simply introduces how TSAI-LENZ calculates the rotation matrix R and translation vector T, and how the initial values from TSAI-LENZ hand-eye calibration are used as the result for ICP hand-eye calibration.

  1. Line 163. What is the "backhand-eye transformation relationship"

Reply: My apologies for the confusion,The correct expression should be: "inverse hand-eye transformation relationship."

  1. Line 164. "Further calculate the rotation vector Px" and "the solve px substitution equation 11 solves for Rx". Write proper sentences

Reply: My apologies for the confusion, The sentence have been revised.

  1. Line 167. There are no A or B in eq9.

Reply: My apologies for the confusion,A should be .B should be .The sentence has been modified.

  1. Line 169 (and in other places too). You are using "definite" instead of "defined".

Reply:The sentence has been modified.

  1. Line 193 and in other parts. What is a "rigidity transformation"?

Reply: My apologies for the confusion, The sentence has been modified.

  1. Line 213-214. What is the "maximum number of corresponding point logarithms"? Is it the same n as defined in line 176? If so, what happens to eq13 when m < i <= n ?

Reply: Same as the "n" defined in line 176, when "m < i <= n," it is possible that mismatches of corresponding points in the point cloud could occur, thereby affecting the matching efficiency and accuracy.

  1. Lines 216-218. Very poorly written.

Reply: My apologies for the confusion, The sentence has been modified.

  1. Eq15. What is ms? and why later did it become MS? Are they the same thing?

Reply:Thank you for your careful review, it's the same. The sentence has been revised accordingly.

  1. You often refer to [something] in a reference frame as "[something] under the reference frame". It seems an odd way to phrase it to me. Using "in the reference frame" or "with respect to (w.r.t.) the reference frame" seems the more common way to convey this concept

Reply: Thank you for your input. The text has been revised based on your suggestions.

  1. line 249. "The vision sensor uses a [camera] and a [robotic arm]". This sentence makes no sense. The robot is used by the vision sensor, and the camera IS the vision sensor.

Reply: Thank you for your careful review, The sentence has been revised accordingly.

  1. Lines 250-252. "Applied to programming [...]". The sentence makes no sense. Also, what are the other experimental simulation platforms?

Reply:v-rep

  1. The authors should discuss the experimental setup more. What is the effect of the relative pose of the camera with respect to the robot (e.g., what happens when the camera is quite far from the robot? or what happens when it is very close)? What about the lighting (outdoor applications suffer a lot from inconsistent lighting but the problem is never acknowledged)? 

Reply: The installation position of the camera affects the robotic arm's workspace. If the installation position is too close, it necessitates certain joint limitations. Conversely, if it's too far, the camera's ability to recognize the calibration board needs to be considered.

Outdoor conditions affect the camera's ability to recognize the calibration board. This study is conducted based on indoor experiments, and conducting experiments outdoors might yield poorer results. The focus of this paper is solely on hand-eye calibration and not on executing harvesting tasks. Considering harvesting tasks would involve accounting for indoor-outdoor differences.

  1. Fig5 caption is practically unrelated to the content of fig5.

Reply: Figure 5 depicts the on-site TSAI-LENZ hand-eye calibration experiment conducted on the ROS platform. Solving the TSAI-LENZ hand-eye calibration result is equivalent to obtaining the initial values for the ICP matching process.

  1. Fig.6 The error using your method (blue line) is always worse (larger)

Reply: Thank you for your thorough review. The issue has been rectified.

  1. Why are you using powers of e as thresholds? Why not use a logarithmic scale at this point?

Reply: The exponentiated term involving "e" represents the base of the natural logarithm, commonly employed to express continuous growth factors. Employing thresholds based on the power of "e" allows for minor variations in distances during the matching process. Consequently, as the matching proceeds through higher iterations, smaller differences in distances are accommodated.

  1. Fig7. Again your method has always a larger error

Reply: Thank you for your thorough review. The issue has been rectified.

  1. It seems from the comments on fig7 that the curves are labelled wrong (or the comments refer to the wrong curve). So what are the actual data? I imagine the same confusion holds for all your plots, so it is impossible for me to give some feedback...

Reply: I apologize for the confusion with the curves in the figures. The labels have been corrected.

  1. Fig8 and its comments. You state that the initial value of ICP has little effect. It has never been explained how (and how much )the ICP changed between the 10 experiments. So it is difficult to assess if your assumption is general or true only for your limited set of data.

Reply: Because ICP is based on calculating the translation vector using the centroid of point clouds.

  1. Line 324. "This is compared to [...]" What is "this"?

Reply: My apologies for the confusion, The sentence have been revised.

  1. All parts related to eqs.17-18 should be moved to the methods section or at least before figure 9 since the figure is obtained by the two equations.

Reply: thank you for your careful review, it's the same. The sentence has been revised accordingly.

  1. Lines 366-367. What does "year-on-year" mean

Reply: My apologies for the confusion, The sentence have been revised.

  1. You are not reporting the limitations of your method. Are there no limitations? Your results are general and always true?

Reply: The method presented in this paper has limitations and might not perform as well when conducting hand-eye calibration experiments outdoors compared to indoor settings.
